# Changes in Environment and Management Practices Improve Foot Health in Zoo-Housed Flamingos

**DOI:** 10.3390/ani13152483

**Published:** 2023-08-01

**Authors:** Andrew Mooney, Kelly McCall, Scott Bastow, Paul Rose

**Affiliations:** 1Dublin Zoo, Phoenix Park, D08 AC98 Dublin, Ireland; 2School of Biological Sciences, Queen’s University Belfast, Belfast BT9 5DL, UK; 3Centre for Research in Animal Behaviour, Psychology, University of Exeter, Exeter EX4 4QG, UK; 4Wildfowl & Wetlands Trust Ltd. (WWT), Slimbridge Wetland Centre, Gloucestershire GL2 7BT, UK

**Keywords:** enclosure design, welfare, bumblefoot, pododermatitis, zoo enclosure

## Abstract

**Simple Summary:**

Flamingos are one of the most commonly kept species of bird in zoos around the world, yet despite this popularity, many management challenges remain, with foot health still a pervasive problem in zoo-housed flamingos. This represents a prominent welfare concern, and has been directly linked to age, climate, and substrate. Foot health has become a particular concern over recent years as zoo-housed birds are spending longer periods indoors due to mandatory indoor housing orders imposed by national governments in response to bird flu. Tracking changes in foot health for a flock of Chilean Flamingos at Dublin Zoo (Ireland) over an 18-month period, we show that providing access to outdoor habitats and natural substrates can improve the health and wellbeing of zoo-housed flamingos. This study highlights the importance of regular foot health monitoring in flamingos, and the importance of natural substrates when designing flamingo habitats.

**Abstract:**

Foot lesions are a highly prevalent phenomenon among zoo-housed flamingos, with up to 99.8% of birds affected. These lesions are a recognized welfare concern, increasing the likelihood of bacterial infections, and even septicemia. Although several risk factors have been linked to foot lesions in flamingos (including age, climate, and substrate), there have been few studies looking at changes in foot lesions over time. This study tracked changes in foot lesions for an individual flock of Chilean Flamingos (97 birds) at Dublin Zoo, Ireland, over an 18-month period in response to a mandatory indoor housing order imposed by the Irish Government as a seasonal precautionary measure to prevent the spread of avian influenza. Using a pre-defined scoring system for four common types of foot lesions (hyperkeratosis, fissures, nodular lesions, and papillomatous growths), we show that providing unrestricted access to outdoor habitats and natural substrates (both terrestrial and aquatic) can improve the health and wellbeing of zoo-housed flamingos. This longitudinal study highlights the importance of regular foot health monitoring in flamingos, and the importance of natural aquatic substrates when managing flamingos. As many zoo-housed birds have been spending more time indoors on artificial substrates over recent years due to avian influenza housing orders, it is critical that we assess the impact of such changes in management and habitat access on bird health and welfare.

## 1. Introduction

A key predictor of health and wellbeing for many species under human care is good foot health, especially for terrestrial species where movement and resting relies on standing or perching. Pododermatitis (often termed bumblefoot) is a foot pathology commonly seen in rodents, lagomorphs, and many different species of bird [1]. Pododermatitis can present as a range of mild to severe clinical signs, and can result in secondary infection that can prove fatal [2]. Abrasion, swelling, and ulceration of the plantar metatarsal and/or digital pads of the foot are key diagnostic signs of pododermatitis, and these pathological changes can lead to infection of the foot’s internal tissues [3]. One or both feet can be affected to equal or varying degrees of severity [4]. Pododermatitis is predominantly a disease of animals under human care, as reports of bumblefoot in free-living birds are very rare or completely absent [5]. Many factors have been suggested as potential causes of pododermatitis, which can predispose animals under human care to the condition. These range from a lack of exercise and opportunities for foot manipulations, to the animals’ mass and the overall hygiene of their habitat [6,7].

Research on agricultural birds has shown the importance of substrate quality and litter cleanliness for the maintenance of good foot health [8]. Lameness in poultry is commonly attributed to bumblefoot [9], and it has been shown that wet litter can increase the chance of pododermatitis developing in domestic turkeys (*Meleagris gallopavo*) [10]. In a zoo setting, substrate also has a role to play in the development of pododermatitis. Studies and observations of raptorial birds, penguins (Sphenisciformes), and waterbirds, such as flamingos (Phoenicopteriformes), have all noted the importance of providing natural substrates and perching opportunities for the maintenance of foot health [11,12,13,14]. For example, Wyss et al. [15] identified that flamingos with access to ponds with a natural substrate presented with significantly less severe foot lesions than birds that used concrete-floored ponds. The same authors also screened the feet of free-living flamingos for pododermatitis and found no evidence of the condition.

The challenge of maintaining healthy feet in flamingo flocks under human care is frequently recorded in the scientific literature [16], and pododermatitis is considered a global problem for those managing this iconic zoo-housed bird [17]. Alongside substrate material, the body mass and age of a flamingo both influence the onset and progression of pododermatitis [15], with older and heavier flamingos presenting with higher prevalence of ulcerative lesions. There is some conjecture that flight restraint (pinioning or feather trimming) may also predispose waterbirds, including flamingos, to more severe foot lesions in comparison to full-winged flamingos which live in covered aviaries. For example, degenerative joint disease (DJD) in zoo-housed wildfowl (Anseriformes) may be more common in birds that have limited opportunities for exercise and reduced access to variable and natural forms of perching [18]. DJD can alter a bird’s weight bearing, and such a change is another potential predisposing factor to the development of pododermatitis. Other authorities have, conversely, speculated that pinioning does not predispose flamingos to more severe pododermatitis [19], and therefore, currently, no definitive conclusion between pododermatitis risk and bird wing condition can be drawn.

Classification of foot lesions in zoo-housed flamingos is provided by Nielsen et al. [20] with four types of pathological lesion noted (Figure 1). Hyperkeratosis (ranging from slight overgrowth to marked overgrowth of skin); fissures (ranging from superficial to deep cracks in the surface of the foot); nodular lesions (those without ulceration to active, ulcerated nodules); and papillomatous growths (finger-like growths that protrude from the edges of each digit). Each foot is scored in turn for the presence of these pathological lesions, taking into account the integrity of each digit and the heel of the foot [20]. Although this methodology assesses each foot lesion independently, they reflect a continuum of disease progression, with hyperkeratosis potentially developing into other, more severe, lesion types [20]. This same study found that foot lesions are a pervasive problem in zoo-housed flamingos globally (854 flamingos across 20 zoological collections), with prevalence rates of 100%, 87%, 17%, and 46% for hyperkeratosis, fissures, nodular lesions, and papillomatous growths, respectively [20]. This statistic is alarming given that flamingos are one of the most commonly kept species of bird in zoos around the world, numbering in the tens of thousands [21].

The key contributing factors to the development of pododermatitis in zoo-housed flamingos are, in general, bird mass and age, the quality and hygiene of the substrate, a sedentary zoo lifestyle, and lack of exercise [12]. However, poor nutrition, behavioral abnormalities, and incorrect weight bearing can also cause the onset of pododermatitis [1,22]. Additionally, the amount of time spent indoors, and climatic conditions, can also play a role in the development of specific types of foot lesions, especially in zoo-housed flamingos [23]. For example, flamingos which spend longer periods of time indoors (and on artificial substrates), and those housed in zoos at higher latitudes which experience lower mean annual temperatures, are significantly more likely to present with pododermatitis lesions [23].

Although several risk factors have been shown to predispose flamingos under human care to foot lesions (e.g., age, climate, and substrate), there have been few studies looking at changes in foot lesions over time in response to changes in management and environmental conditions. This study tracked individual changes in the prevalence and severity of foot lesions for a flock of 97 Chilean Flamingos (*Phoenicopterus chilensis*) at Dublin Zoo, Ireland, in response to mandatory environmental and management changes over an 18-month period. The foot condition, and the degree of pododermatitis presented, were evaluated for each individual before, during, and after a period of mandatory indoor housing imposed by the Irish Government as a seasonal precautionary measure to help prevent the spread of avian influenza (AI) [24]. This study provides evidence for necessary alterations to indoor habitats and husbandry practices to promote flamingo wellbeing during mandatory AI housing orders. Simultaneously, this paper also aims to explore the current methodology used to score flamingo foot lesions, while also discussing the outcomes from a practical animal management and welfare perspective.

## 2. Materials and Methods

We quantified and compared temporal trends in flamingo foot lesions for 97 Chilean Flamingos at Dublin Zoo between May 2021 and November 2022 in response to changes in management and environmental conditions as a result of mandatory AI housing restrictions.

### 2.1. Flamingo Management at Dublin Zoo

The Chilean Flamingos at Dublin Zoo are housed in a large walk-through outdoor aviary (approximately 1350 m^2^; Figure 2a). This aviary comprises a land mass (approximately 600 m^2^) and a shallow lake (~1 m depth) which flows through the habitat providing opportunities for the flamingos to swim and forage. The outdoor habitat has a sand substrate throughout (both terrestrial and aquatic) in addition to a nesting area with clay. The outdoor habitat also includes a small artificial concrete feeding pool, and natural vegetation to provide cover. The indoor habitat comprises two connected rooms. The first room (47 m^2^; Figure 2b) consists of a completely textured rubber floor with a surrounding sprinkler system to keep the floor wet when needed. The room is completely enclosed, with four translucent roof lights to provide natural lighting. This is connected to a second room (60 m^2^; Figure 2c) which consists of three mesh walls open to the outside air. One of these mesh walls is covered in plastic lining, and the roof is covered with aluminum sheets to provide cover. This is a dry room and contains a rounded pebble substrate (0.5–1 cm diameter) mixed with sand. Throughout the year the flamingos are given 24-h access to both the second room of their indoor habitat (60 m^2^) and outdoor habitat, except in rare cases of extremely cold or icy weather, in which case they are kept completely indoors. When indoors, flamingos are fed from water-filled plastic trays, which are distributed around the first indoor room. The rubber flooring of the first indoor room is cleaned and disinfected daily, with the birds given immediate access to the room once cleaning has been completed. The second indoor room, consisting of a natural substrate and open to the outside air, is not disinfected daily and is not used for feeding.

On 6 May 2021 (Time Point A), all flamingos were living in their outdoor habitat and underwent routine veterinary health checks, including both ringing and microchipping of individual birds. At this time, photographs were taken of each flamingo’s feet as part of a long-term foot health monitoring project being undertaken at Dublin Zoo. This was undertaken in a standardized way for each flamingo, with both feet placed against a solid color board to increase the clarity and discernability of foot lesions. Both the date and individual animal identification number were also written on the board to minimize any reporting errors. Photographs were then taken of both feet using a mobile device camera (12-megapixels; iPhone 13 Pro Max, Apple Inc., Cupertino, CA, USA).

On 1 December 2021, all flamingos were brought into their indoor habitat as part of a national confinement order for poultry and captive birds imposed by the Department of Agriculture Food and the Marine, as a precautionary measure against the emerging avian influenza (S.I. No. 607/2021—Avian Influenza (Precautionary Confinement of Birds) Regulations 2021) [24]. This order was in addition to the robust ‘Avian Influenza Biosecurity Plan’ already in operation at Dublin Zoo, with each bird habitat managed as a separate epidemiological unit and regular staff biosecurity training. This legal confinement order continued until April 2022 [24]. The flamingos were then provided access to their outdoor habitat again on 16 April 2022 (Time Point B). At the time of release all flamingos were vaccinated against AI and given routine veterinary health checks. Photographs were again taken of each flamingo’s feet at this time. On 9 November 2022 (Time Point C), having been outside for more than six months, the flamingos were given a further dosage of the AI vaccine. Photographs were again taken of each flamingo’s feet at this time as part of routine veterinary monitoring.

These photographs provide a comprehensive dataset to assess the foot health of each flamingo in response to environmental and management changes, with data available for each flamingo before confinement (A), at the end of confinement (B), and over six months post confinement (C).

### 2.2. Flamingo Foot Scoring

All flamingo foot photographs for each time point (A, B and C; *n* = 97 each) were assessed independently by two trained evaluators (K. M. and S. B.) using the foot lesion classification and scoring system developed by Nielsen et al. [20]. Both evaluators were trained by providing alternative and independent sets of practice photographs prior to the scoring of the current study photographs.

This system divides the plantar surface of the foot into seven discrete parts, considering both the integrity of each digit and the heel of the foot. Each of the seven parts of the foot were then scored for the four types of foot lesion (hyperkeratosis, fissures, nodular lesions, and papillomatous growths), with each part assigned a value from 0 to 2 depending on the severity of the foot lesion type present. A score of 0 indicates no lesion of that type was present. All foot lesion scores for each lesion type were then summed to create an overall score per foot per lesion type, and subsequently per flamingo, for each time point. All foot lesion scores were then combined to generate an overall foot lesion score per foot, and per flamingo, for each time point (considering all lesion types).

As there have been doubts raised over the reliability of the scoring system developed by Nielsen et al. [20], in terms of both inter- and intra-evaluator agreement (depending on the type of foot lesion) [25], the independently calculated foot scores were compared for each evaluator. In instances where the foot scores did not match, a consensus was sought between both evaluators to provide a final value for subsequent analysis. If instances arose where a consensus could not be reached by both evaluators, then a third evaluator (A.M.) was consulted to provide a final score for subsequent analysis. Discrepancies in the total foot scores between both evaluators were subsequently quantified to provide an estimate of the inter-evaluator reliability of Nielsen et al.’s [20] scoring system.

### 2.3. Data Analysis

All data screening and cleaning, and subsequent analyses, were carried out using the R program (version 1.4.1106) [26]. Descriptive statistics for both prevalence and foot lesion severity (mean foot score ± SD) were generated for each time point. Changes in overall foot scores for each flamingo (considering all lesion types) between the three time points (A, B, and C) were assessed using a Friedman Test. Assumptions of normality were assessed visually using Q-Q plots and histograms, and assumptions of sphericity were assessed using Mauchly’s sphericity test. Differences between the overall foot lesion score per foot of each flamingo (i.e., left or right foot) were also assessed separately for each time point (A, B, and C) using three separate paired Wilcoxon Signed-Rank Tests with a Bonferroni adjustment (considering all lesion types). To assess whether sex or age influenced how individual flamingos responded to changes in their habitat, we modeled the change in overall foot lesion score observed per flamingo (delta; Δ) between time points (A and B, and then B and C) using two separate generalized linear models with sex and age included as explanatory variables. Model selection was undertaken using AIC values on all subsets of the maximal model (i.e., both sex and age as explanatory variables). A threshold of more than two AIC units lower than the nearest competing model was considered sufficient for final model selection [27]. Residual diagnostic plots were generated to confirm the validity of both final models. Correlations between sex, age, and the severity of each individual lesion type were also assessed for each time point using the Spearman correlation.

## 3. Results

### 3.1. Foot Lesion Prevelance and Severity

All flamingos included in this study (*n* = 97) suffered from at least one type of foot lesion at some point in time. The prevalence of foot lesions was very high at both Time Points A and B (99.0% and 100% of flamingos, respectively); however, this dropped considerably at Time Point C (59.8% of flamingos) (Table 1). Hyperkeratosis was the most commonly identified foot lesion at both Time Points A and C (99.0% and 53.6% of flamingos, respectively), whereas papillomatous growths were the most commonly identified foot lesion at Time Point B (100% of birds). The prevalence of nodular lesions was consistently low, ranging from 0–4.1% of flamingos, whereas the prevalence of fissures ranged from 12.4% to 70.1% of flamingos (Table 1).

Although the prevalence of foot lesions was high during both Time Points A and B, the mean foot score was much higher for Time Point B (13.75 ± 6.81) compared to Time Point A (8.60 ± 3.83) (Table 1), suggesting a decrease in overall foot heath between Time Points A and B. In comparison, Time Point C had the lowest mean foot score for any period assessed (1.78 ± 2.54), suggesting an improvement in foot health between Time Points B and C. Papillomatous growths accounted for the highest foot scores of any lesion type during both Time Points A and B (4.08 ± 2.84 and 8.68 ± 4.86, respectively), whereas hyperkeratosis accounted for the highest score at Time Point C (1.05 ± 1.39). In contrast, nodular lesions and fissures had much lower foot scores, ranging from 0 to 0.05 ± 0.27, and 0.19 ± 0.63 to 2.24 ± 2.61, respectively (Table 1). No significant difference was observed between the overall foot lesion score per foot of each flamingo (i.e., left or right foot), with *p* > 0.05 (Wilcoxon Signed-Rank Tests) observed for each time point (A, B, and C).

### 3.2. Response to Changes in Management and Environmental Conditions

The overall foot lesion score per flamingo was statistically significantly different at each of the three time points (Friedman test, X^2^(2) = 159.52, *p* < 0.0001). Post hoc analyses (pairwise Wilcoxon Signed-Rank Tests with a Bonferroni adjustment) revealed that all pairwise comparisons between Time Points A, B, and C were statistically significantly different (*p* < 0.0001 for all) (Figure 3). These results confirm a significant decrease in overall foot heath between Time Points A and B (following the movement of the flamingos indoors), and a significant recovery in foot health between Time Points B and C (following the release of the flamingos into their outdoor habitat). However, both Time Points A and C were also statistically significantly different, despite the flamingos having access to their outdoor habitat at both time points.

### 3.3. Age and Sex

Both age and sex were assessed as possible predictors of the change in overall foot lesion score observed per flamingo (delta; Δ) between time points (A and B, and then B and C). No effect of sex was observed, and in both instances the variable was removed from the model during the model selection process (using AIC values) (Figure 4). Similarly, there was no observed effect of age on the change in individual foot score between Time Points A and B (*p* > 0.05), suggesting all flamingos were equally as susceptible to foot lesions and decreases in foot health regardless of their age or sex. However, a positive relationship was found between age and the change in overall foot lesion score observed per flamingo between Time Points B and C (*p* < 0.05). In this case a positive relationship is reflective of a smaller change in overall foot lesion score (delta; Δ) between time points (Figure 5). Therefore, this implies that the feet of older flamingos did not recover as quickly as those of younger flamingos.

In terms of correlations between sex, age, and individual lesion types, we found a significant correlation between sex and papillomatous growths at Time Point B (*p* < 0.05; Figure 6), with male flamingos more likely to have higher scores for papillomatous growths when indoors compared to females. We also found a significant negative correlation between age and papillomatous growth scores at Time Point B (*p* < 0.01; Figure 6), suggesting older flamingos developed less severe papillomatous growth infections when indoors compared to younger flamingos. Similarly, a significant positive correlation was found between age and hyperkeratosis at Time Point C (*p* < 0.001; Figure 6), suggesting older flamingos did not recover as quickly as younger flamingos, and had more severe hyperkeratosis infections after being outside for six months.

### 3.4. Inter-Evaluator Reliability

Discrepancies in the total foot scores generated by both independent evaluators were minimal, with a mean difference in total foot scores of only 0.60 ± 0.97. Of the total number of foot pairs scored (291), 60.1% had complete agreement and a further 28.6% were within one point of each other. These findings suggest that although not perfect, the scoring system developed by Nielsen et al. [20] can be regarded as somewhat reliable when utilized by trained evaluators. Time Point C had the greatest agreement, with 81.2% of scores in complete agreement, whereas Time Point B had the lowest agreement, with only 42.2% of scores in complete agreement. These time points coincide with the highest (Time Point B) and lowest (Time Point C) mean foot scores, suggesting that as foot health decreases, and scores increase, the scoring system becomes less reliable. Of the four lesion types assessed, papillomatous growths showed the greatest discrepancy, with a mean difference in foot scores of 0.43 ± 0.99. In contrast, the evaluators showed much greater agreement when scoring nodular lesions, with a mean difference in foot scores of 0.003 ± 0.57. In comparison, hyperkeratosis had a mean difference in foot scores of 0.28 ± 0.53, and fissures had a mean difference in foot scores of 0.15 ± 0.42. Consensus was sought for all scores where a discrepancy was found, and all analyses presented above are based on these consensus foot scores.

## 4. Discussion

Results from this study show significant changes in flamingo foot health following a mandatory indoor housing order imposed by the Irish Government in response to AI. We found that flamingo foot health decreased significantly when animals were moved indoors onto an artificial and dry pebble-sand substrate. However, we also reveal an extremely rapid recovery in individual flamingo foot health once the flamingos were given access to their outdoor habitat and a natural substrate (both terrestrial and aquatic). However, important factors, such as the age of individual animals, were also identified, and should be taken into consideration when managing flamingos and assessing foot health.

Consistent with previous findings, we confirm that housing flamingos on artificial substrates can increase the prevalence of foot lesions [23]. We found that the highest prevalence and severity of foot lesions occurred after the flamingos had been housed indoors on a rubber and dry pebble-sand substrate for six months (Time Point B). At this time, 100% of birds suffered from papillomatous growths and 70.1% suffered from fissures. In comparison, the prevalence of these foot lesions dropped significantly to 22.7% and 12.4%, respectively, after the flamingos had been given access to their outdoor habitat and natural substrates. This mirrors previous findings highlighting the importance of natural aquatic substrates for the maintenance of good flamingo foot health [15]. Although the prevalence (and severity) of hyperkeratosis also decreased over the same period, from 93.8% to 53.6%, most birds were still suffering from hyperkeratosis six months after they had been given access to their outdoor habitat. This suggests that either hyperkeratosis has a longer recovery time compared to other lesion types, or that other factors are influencing the development and persistence of hyperkeratosis. The prevalence of nodular lesions was incredibly low across all time points, contrary to the results of previous studies [20,28]. Interestingly, both the prevalence and severity of nodular lesions were lowest during Time Point B. This is consistent with previous findings showing that nodular lesions are less affected by indoor housing than other lesion types [23].

Surprisingly, a significant difference was found between the overall foot lesion score per flamingo at Time Points A and C, both of which coincide with when the flamingos had access to their outdoor habitat and a natural substrate. However, it should be noted that the flamingos had previously been restricted to their indoor habitat as part of an earlier national confinement order to prevent the spread of AI. This confinement order lasted from December 2020 to April 2021, and therefore the foot scores at Time Point A (May 2021) reflect feet in the process of recovering. This may also explain why Time Point A showed the most prevalent and severe hyperkeratosis scores of any Time Point, as hyperkeratosis may have a prolonged recovery period. The decrease in hyperkeratosis observed between Time Points A and B could also be explained by the potential development of hyperkeratosis into further, more severe, lesion types, as has been recorded in other species [20,29]. This can be seen as further evidence of the role that housing and substrate play in the maintenance of good flamingo foot health in zoos, and the difficulties encountered when trying to effectively monitor and measure foot health.

The relatively small discrepancies observed in the overall inter-evaluator reliability (88.7% of scores within one point of each other) suggest that the classification method developed by Nielsen et al. [20] can be considered a useful tool for categorizing and rating flamingo foot lesions. Despite previous findings of varying inter-observer reliability using the same scoring method [25], the current study showed reasonably consistent scores, which may be explained by prior training of evaluators using alternative and independent sets of practice photographs prior to the scoring of the current study photographs. Reliability was highest when foot lesions were least severe (Time Point C) and lowest when foot lesions were most severe (Time Point B), suggesting that the Nielsen et al. [20] scoring method becomes less reliable as lesion severity increases. While scoring based on photographs may not allow for the most accurate measurement of lesions [20], we find that it still provides a relatively quick, convenient, and practical method of evaluating foot health for a large number of animals simultaneously. However, we do believe that it could be further improved by providing a more expansive set of reference materials (i.e., example photographs), and a more standardized and discrete classification system for both prevalence and severity of each lesion type.

Contrary to other research, we found no significant link between the sex of an individual and their change in overall foot lesion score over time [15,28]. However, male flamingos had significantly higher papillomatous growth scores at Time Point B compared to females. Similarly, we found no significant effect of age on the change in individual overall foot score between Time Points A and B. This suggests that the age of an animal does not influence their susceptibility to foot lesions overall, or the rate of foot lesion development. The only exception to this was for papillomatous growths, where foot lesion severity at Time Point B was higher for younger individuals compared to older individuals. Additionally, a significant relationship was found between the age of an individual and the change in overall foot lesion score between Time Points B and C. This suggests that older flamingos do not recover as quickly as younger flamingos. This is an important factor to consider from a management and veterinary perspective, as foot lesions can lead to complications such as septicaemia [30]. Providing sufficient time for feet to fully recover is therefore an important consideration if flamingos are going to be repeatedly placed under AI housing orders. In addition to substrate, several other factors have previously been shown to influence flamingo foot health, including diet, bird mass, and climate (latitude and temperatures) [17,23], and future scoring of flamingo feet should attempt to include such explanatory factors in any analyses.

The health and welfare of zoo-housed animals is a growing concern and requires a sufficient evidence-base to inform effective and sustainable management decisions [31]. Through this study we add to the existing knowledge which can be used to inform the management of zoo-housed flamingos. The results presented here highlight the benefits of multiple measurements of specific health parameters, allowing for a wider understanding of how individual flamingos respond to a variety of management conditions and how they can recover when provided with appropriate conditions. Ultimately, housing flamingos indoors, and on artificial substrates, has a negative impact on long-term foot health. Therefore, we encourage zoos to provide flamingos with unrestricted access to outdoor habitats and natural substrates, both terrestrial and aquatic, wherever possible to ensure good flamingo foot health [11,17]. Although costly and logistically difficult, the provision of natural substrates indoors, particularly natural aquatic substrates indoors, should also be explored as a way of minimising the impact of indoor housing orders on flamingo foot health and welfare. Keeping flamingos indoors for prolonged periods of time should only be undertaken when absolutely necessary and after other management options have been exhausted. With increasingly regular indoor housing orders to protect birds from global zoonoses, this may be a necessary step to ensure the long-term health and welfare of zoo-housed flamingos. Conversely, it can also be argued that vaccination, and increased biosecurity of outdoor habitats where possible, are more practical and effective long-term solutions for zoo-housed animals compared to increasingly regular and mandatory indoor housing orders [32]. Mandatory indoor housing orders are now a global issue, which impact thousands of zoo-housed birds annually, not just flamingos [21,33]. Therefore, it is critical that we find effective solutions to both protect animals from emerging diseases (such as AI), while also considering the welfare of individual animals [34].

## 5. Conclusions

We have shown that up to 100% of the zoo-housed flamingos in this study were impacted by some form of foot lesion, with the type and severity changing with management practices and environmental conditions. A significant decrease in foot health was observed when flamingos were housed indoors on a rubber and dry pebble-sand substrate, without access to natural aquatic substrates (increase in mean foot score from 8.60 to 13.75). Conversely, foot health significantly increased when the flamingos were provided with access to an outdoor habitat with natural substrates, both terrestrial and aquatic (decrease in mean foot score from 13.75 to 2.54). We recommend that zoo-housed flamingos be provided with outdoor habitats and natural aquatic substrates wherever possible. Although mandatory indoor housing orders are likely to continue in the near future, the impacts of such actions may potentially be mitigated against by providing natural substrates indoors, both terrestrial and aquatic. Additionally, age-dependent recovery times, and prolonged recovery times for hyperkeratosis, both need to be considered when monitoring and evaluating flamingo foot health and welfare.

## Figures and Tables

**Figure 1 animals-13-02483-f001:**
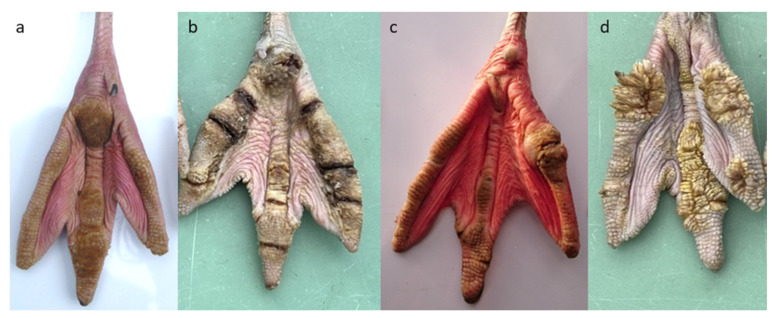
Classification of flamingo foot lesions according to Nielsen et al. [20]. From left: (**a**) hyperkeratosis, (**b**) fissures, (**c**) nodular lesions, (**d**) papillomatous growths. Photos: Paul Rose (**a**,**c**) and Andrew Mooney (**b**,**d**).

**Figure 2 animals-13-02483-f002:**
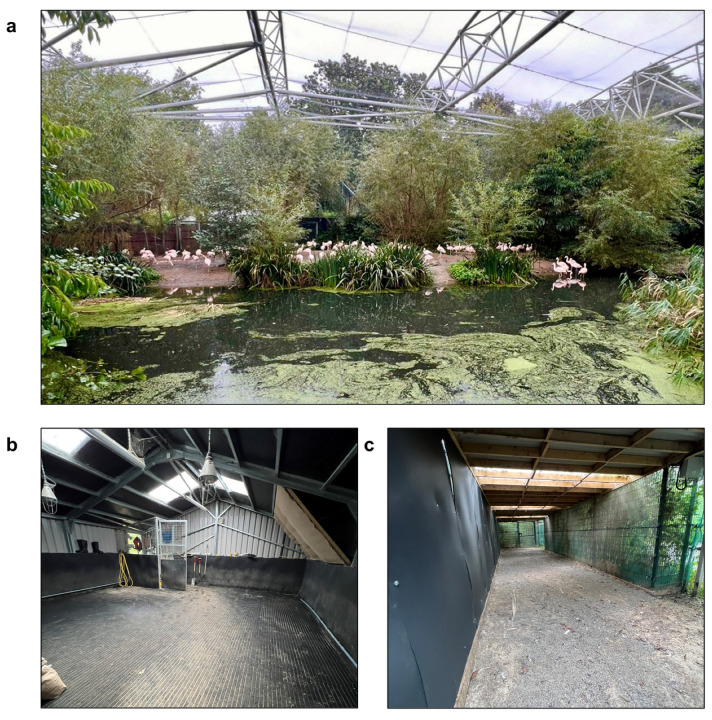
The Chilean Flamingo habitat at Dublin Zoo. This includes a 1350 m^2^ outdoor aviary, comprising a land mass and a shallow lake surrounded by natural vegetation (**a**). The indoor habitat comprises two connected rooms. The first room is 47 m^2^ and consists of a completely textured rubber floor (**b**). This is connected to a second 60 m^2^ room which has a pebble and sand substrate, and mesh walls which are open to the outside air (**c**). Photos: Andrew Mooney.

**Figure 3 animals-13-02483-f003:**
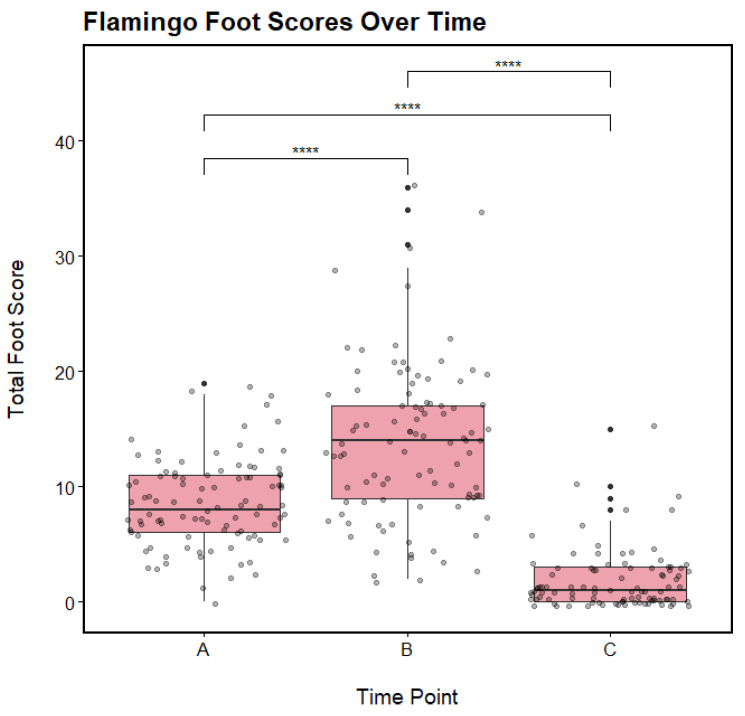
The total foot scores for 97 Chilean Flamingos at Dublin Zoo at three time points (A = 6 May 2021, B = 16 April 2022, and C = 9 November 2022). These scores reflect the scoring metric developed by Nielsen et al. [20] and include the four types of common flamingo foot lesion (hyperkeratosis, fissures, nodular lesions, and papillomatous growths). Pink boxes denote interquartile ranges (25th to 75th percentiles), central horizontal lines denote median values (50th percentile), and black whiskers denote 5th and 95th percentiles. Solid black dots denote outliers, and shaded black dots show the distribution of the data. *p* ≤ 0.0001 for all pairwise comparisons (****).

**Figure 4 animals-13-02483-f004:**
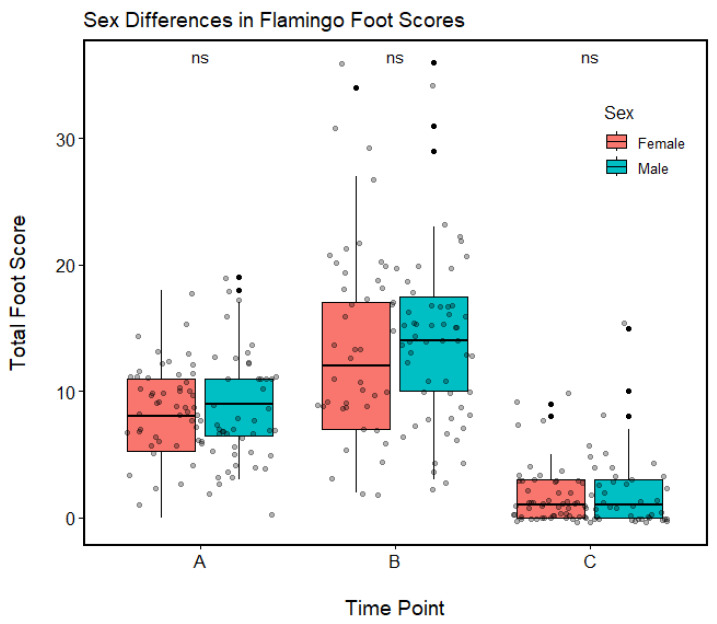
The foot scores for the 42 female and 55 male Chilean Flamingos at Dublin Zoo at three time points (A = 6 May 2021, B = 16 April 2022, and C = 9 November 2022). These scores reflect the scoring metric developed by Nielsen et al. [20] and include the four types of common flamingo foot lesion (hyperkeratosis, fissures, nodular lesions, and papillomatous growths). All comparisons between males and females at each time point (A, B, and C) are non-significant (ns). Pink and blue boxes denote interquartile ranges (25th to 75th percentiles), central horizontal lines denote median values (50th percentiles), and black whiskers denote 5th and 95th percentiles. Solid black dots denote outliers, and shaded black dots show the distribution of the data.

**Figure 5 animals-13-02483-f005:**
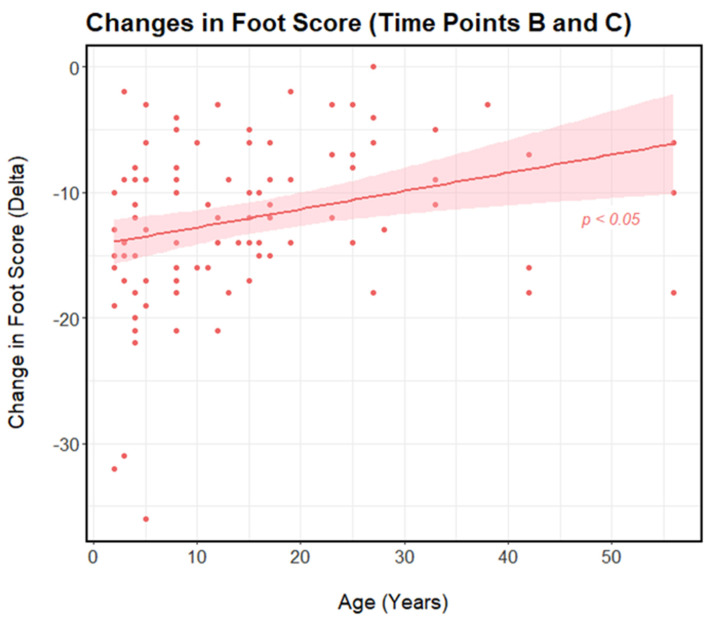
The relationship between age and the change in overall foot lesion score observed (delta; Δ) per flamingo between Time Points B (16 April 2022) and C (9 November 2022) (*n* = 97). Here, a reduction in foot score reflects an improvement in individual flamingo foot health. Foot scores reflect the scoring metric developed by Nielsen et al. [20] and include the four types of common flamingo foot lesion (hyperkeratosis, fissures, nodular lesions, and papillomatous growths). The solid pink line represents a linear regression line, the shaded pink areas around the line represent 95% confidence intervals, and solid pink dots show the distribution of the data.

**Figure 6 animals-13-02483-f006:**
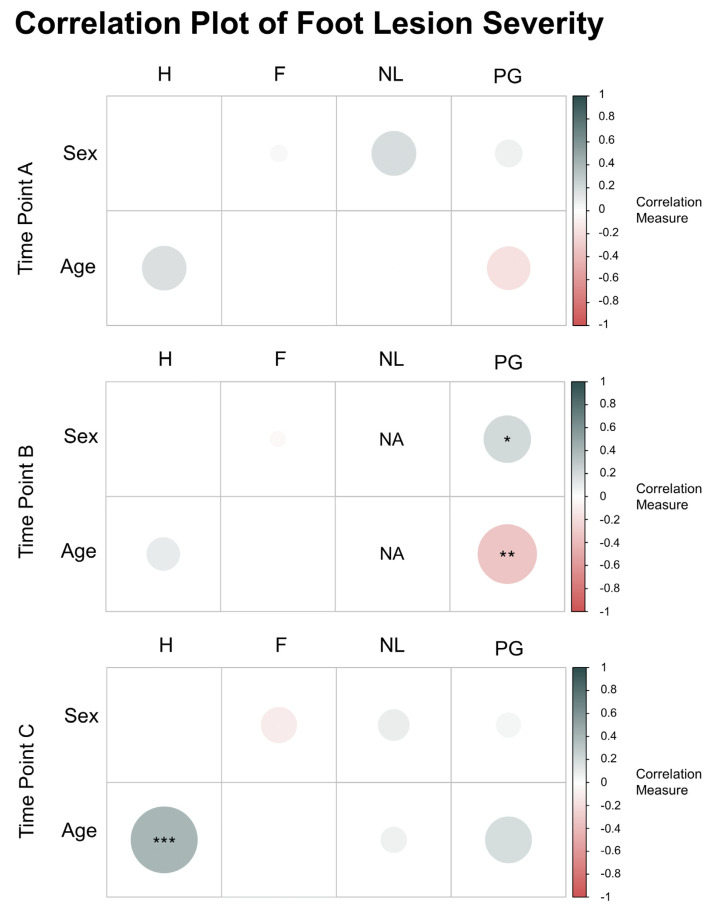
Correlations between sex, age, and the severity of each individual lesion type (H = hyperkeratosis, F = fissures, NL = nodular lesions, and PG = papillomatous growths) for Time Points A (6 May 2021), B (16 April 2022), and C (9 November 2022). (*n* = 97 each). Correlation measures reflect Spearman’s rank correlation coefficient. Scores for each lesion type reflect the scoring metric developed by Nielsen et al. [20]. Significance values: *p* < 0.001 (***), *p* < 0.01 (**), and *p* < 0.05 (*).

**Table 1 animals-13-02483-t001:** The prevalence of foot lesions (hyperkeratosis, fissures, nodular lesions, and papillomatous growths) and mean (±SD) foot score for each lesion type at Time Points A (6 May 2021), B (16 April 2022), and C (9 November 2022).

	Prevalence		
	A	B	C
Foot Lesions (overall)	99.0% (96/97)	100% (97/97)	59.8% (58/97)
- Hyperkeratosis	99.0% (96/97)	93.8% (91/97)	53.6% (52/97)
- Fissures	58.8% (57/97)	70.1% (68/97)	12.4% (12/97)
- Nodular Lesions	4.1% (4/97)	0% (0/97)	1.0% (1/97)
- Papillomatous Growths	89.7% (87/97)	100% (97/97)	22.7% (22/97)
	**Mean (±SD) Foot Score**		
	**A**	**B**	**C**
Foot Lesions (overall)	8.60 (3.83)	13.75 (6.81)	1.78 (2.54)
- Hyperkeratosis	3.40 (1.93)	2.73 (1.78)	1.05 (1.39)
- Fissures	1.06 (1.20)	2.24 (2.61)	0.19 (0.63)
- Nodular Lesions	0.05 (0.27)	0	0.02 (0.20)
- Papillomatous Growths	4.08 (2.84)	8.68 (4.86)	0.53 (1.40)

## Data Availability

Anonymized data files and all associated metadata are available from Zenodo (https://doi.org/10.5281/zenodo.8020339, accessed on 13 June 2023). In line with the FAIR data principles, the data are made available under the following creative commons license: Attribution 4.0 International (CC BY 4.0).

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
