# Peer review of "Changes in Environment and Management Practices Improve Foot Health in Zoo-Housed Flamingos"

_animals, 2023, doi:10.3390/ani13152483_

Round 1

Reviewer 1 Report

A very interesting manuscript, touching on the current problem that many zoos are struggling with. During the migration of birds, on their migration routes there are zoos, keeping many species of birds in their breeding. Migratory birds pose a potential risk of infection with avian influenza. Currently, in order to reduce the likelihood of an outbreak, birds kept in the zoo are as isolated as possible from the external environment. Such keeping of animals certainly has a negative impact on their welfare.

The authors of the manuscript describe the impact of the period of environmental isolation of Chilean flamingos on their welfare, expressed by the intensity and quality of bird foot lesions.

Skin lesions of the feet were classified, assigned to one of four categories and identified in the same way each time through digital imaging.  This was attempted to determine the effect of the sex and age of animals on the incidence of each of the identified foot diseases.

The abundance and homogeneity of the bird research group is beyond doubt. The observation period was divided into three periods and correctly described.

Necessary additions and corrections:

1.       Out of the duty of the reviewer and the researcher's inquisitiveness, I would like to ask you to clarify and describe the technological cycle of bird handling in the zoo:

·         where and how the animals were fed in the indoor enclosures. Flamingos wade in search of food, and therefore are in constant moderate movement that can affect the "health of the feet" – was this possible in the described group of animals?

·         How often the indoor enclosure ( two rooms)  were cleaned and disinfected, whether the animals had direct contact with the freshly disinfected surface. I think that supplementing the text with this information will contribute to the possibility of testing the applied hypothesis, taking into account direct environmental factors and full repeatability of experiments in other zoos.

2.       The include the application of the Bonferroni amendment (269), which is of course legitimate and correct, but in chapter 2.3. Data Analysis (224) has not been remedied and needs to be supplemented.

I leave to the authors' consideration the possibility of changing:

In chapter 3. Result, I encourage authors to divide table 1 into two parts:  One - 3 (A,B,C) Prevalance charts and Second - Mean Foot Score table. This will visually enrich the manuscript and become more readable in reception.

Enriching the work with more photographs from all research periods will certainly have a positive impact on the citation of the manuscript.

Summarizing

I consider the references in the literature review to be correct, giving a picture of the current state of knowledge in the field presented in the manuscript. A manuscript important for future knowledge in the field of bird breeding in zoos, the obtained results certainly affect the state of knowledge, which positively translates into animal welfare.

After introducing the indicated additions and minor corrections in the text, I recommend the manuscript for printing.

Reviewer 2 Report

This is a very well-written manuscript with some findings that were different from previous studies on the same topic. It contributes to the overall knowledge of flamingo foot health, and adds to current literature. The minor points below should be addressed.

Abstract:

Line 21 - 99.8% of flamingos having foot lesions is based on very data from a limited number of institutions, and including it in the abstract may seem a bit alarmist.

Introduction:

Figure 1 - Is this taken directly from Nielsen et al, or photographs taken by the author and classified according to Nielsen et al? If it is the former, then proper copyright permissions need to be sought.

Reviewer 3 Report

I greatly appreciate the authors' clear and concise writing and experimentation. The manuscript includes a thorough introduction on the prevalence of pododermatitis, including the welfare implications of this condition, which is prevalent among zoo-housed flamingos and other species. The experimental methods are sound and straightforward which lends to being replicable at other institutions. I found only one minor typo on Ln 177, "environmental and management and changes" in which the second "and" should be deleted. Otherwise, I found no errors within the manuscript in the text, nor the experimental design or statistical analyses. I believe the content of this manuscript will be valuable to many people working in zoos because pododermatitis is a prevalent condition.

The main claim of this paper is that outdoor natural substrates are better than indoor substrates for flamingo foot health. While this is often assumed, having concrete evidence of the impacts of environment/substrate on foot health for a large group of individuals (97 flamingos) is significant. Particularly as rises in the risk of avian flu have led many zoos worldwide to house birds indoors for extended periods of time.    This paper stands out from other zoo-based research because it examines a large number of individuals (97 flamingos) over an extended period of time (18 months). Additionally, the paper examines 2 key health concerns for birds: bumblefoot and avian flu. Bumblefoot is a common concern in zoo-housed birds, such as flamingos. Additionally, avain flu is increasingly becoming a threat to zoo-housed bird populations that often lead to lockdowns whereby birds are mandated to stay indoors to mitigate risk of catching avian flu. The authors examine how these lockdowns can impact bird foot health and instances of bumblefoot.   While other work has been published on avian foot health (specifically bumblefoot), this disease remains prevalent in zoo-housed bird populations. The authors add a novel component to this literature by assessing how flamingo foot health was impacted due to environmental changes related to mandatory avian flu lockdowns.   The claims and clear and practical.    The study methodology was thorough in its breadth. Avian flu is increasingly a threat to zoo housed birds, and the authors used these circumstances (i.e., mandatory lockdowns) as a natural experiment to better understand how lockdowns and subsequent environmental changes can impact flamingo foot health.    I believe the authors have conducted a thorough study. The study included tracking the foot health of 97 flamingos over an 18-month period as the birds experienced environmental changes related to mandatory lockdowns due to risk of avian flu.   The authors provide a thorough review of the literature on bumblefoot and its health and welfare impacts on zoo housed animals. 
